# Economical management of virtual power plant source-network-load-storage in the context of electricity-carbon market

**Panhong Zhang**[1]*, **Yilu Zheng**[2]

**1** School of Finance, Hubei University of Economics, Wuhan, China, **2** School of Automation, China University of Geosciences, Wuhan, China

* phongzhang@163.com

## Abstract

Under the imperative of achieving dual-carbon goals, the number of distributed energy resources are gradually increasing, thereby amplifying the challenges to grid stability and power balance. Consequently, there is an urgent need to leverage the potential of source-network-load-storage for enhanced power regulation and control. This paper proposes a cloud-edge-end-based multi-time scale economical management of virtual power plant (VPP) source-network-load-storage in the context of electricity-carbon market. In the first layer, a cloud-edge scheduling approach is used to optimize the source-network-load-storage system of the VPP over a long time horizon, aiming to maximize economic benefits. In the second layer, a novel real-time pricing mechanism is employed to effectively manage and regulate the electric vehicle (EV) storage and charging stations. After obtaining the economic management parameters from the previous layer, the second layer employs a real-time scheduling approach based on end-side model predictive control (MPC), to address multi-energy supply-demand fluctuations. To achieve efficient solution, the original two-layer optimization problem is reformulated using mixed-integer linear programming (MILP). Comparative analyses have demonstrated the superior economic and practical performance of the proposed two-layer optimization approach. Simulation results indicate that the total operating cost of the system can be reduced by 2.35%, with a higher flexibility of electricity market operations.

## 1. Introduction

Against the backdrop of profound energy structure adjustment, the share of renewable energy in the power system has been growing rapidly. The intermittency and uncertainty of new energy sources present significant challenges to the safe and stable operation of the power grid. A virtual power plant (VPP) represents a centralized network comprising decentralized energy resources, including photovoltaic (PV),

**Data availability statement:** All relevant data are within the manuscript.

**Funding:** The author(s) received no specific funding for this work.

**Competing interests:** The authors have declared that no competing interests exist.

wind turbines (WT), energy storage systems (ESS), and flexible demand-side loads. These resources are aggregated and controlled uniformly to mimic the operation of a single, unified power plant. The VPP concept emerged in response to the widespread adoption of renewable energy sources (RESs) and the pressing demand for flexible, interactive, and efficient grid systems [1]. Traditional power grids, often reliant on large, centralized fossil fuel-based power plants, face challenges in handling the RES volatility [2]. VPPs provide a solution by facilitating enhanced coordination of source-network-load-storage systems., making it possible to balance supply and demand.

The motivation behind VPPs is to maximize the use of RESs, enhance grid stability, and achieve the dual carbon goal [3]. By aggregating small-scale energy resources, VPPs are capable of offering network services, including load balancing and frequency regulation., and peak shaving. Moreover, VPPs can improve the economics of renewable energy generation, making it more cost-effective for consumers and reducing reliance on fossil fuels. VPPs also support the transition to a decentralized new energy system, empowering consumers to become active participants in electricity-carbon markets. Therefore, the optimal scheduling of VPPs with source-network-load-storage collaboration has become an important topic.

To address economical management coordination problem of VPP in the context of electricity-carbon market, this paper proposes a multi-time scale VPP source-network-load-storage management framework, based on the cloud-edge-end collaborative technology. The upper layer utilizes cloud-edge collaborative scheduling to optimize long-term VPP operations, incorporating electricity-carbon market dynamics for overall economic efficiency. The lower layer applies an end-side model predictive control (MPC) strategy to manage EV storage and charging stations charging behavior and balance short-term supply and demand, driven by a transaction-based real-time pricing mechanism. This hierarchical approach enables scalable, adaptive, and economically efficient VPP scheduling. This paper makes the following contributions:

1) While the previous literature is mainly limited to the parts of source-network-load-storage, the inherent system availability and dispatchability from other flexible parts cannot be fully exploited. This paper proposes a unified VPP source-network-load-storage management framework, which leverages cloud-edge-end collaborative technologies to achieve the real-time data processing and coordination. This framework enables coordinated and flexible control of distributed generation, distribution networks, EVs, and energy storage systems, thereby reducing both system energy consumption and operational costs.

2) A two-layer cloud-edge-end-based multi-time scale economic management framework for VPP source-network-load-storage is proposed to enable dynamic and hierarchical control of the system. In the first layer, day-ahead economic scheduling of source-network-load-storage resources are performed at the cloud-edge level. In the second layer, a novel real-time supply-demand pricing mechanism is introduced to optimize the scheduling of EV storage and charging stations based on end-side MPC. Compared to a single-layer framework, the two-layer model offers enhanced cost-effectiveness and is easier to implement.

3) While RTP and TOU pricing mechanisms generally ignore the supply–demand behavior, an interactive real-time pricing mechanism is proposed for EV storage and charging stations. This mechanism dynamically adjusts prices based on real-time supply-demand conditions, aiming to better align economic incentives with system needs.

4) To address the non-convexity inherent in the source-network-load-storage and real-time electricity pricing model, a mixed-integer linear programming (MILP) relaxation is employed. This approach ensures the preservation of convexity and enhances the convergence performance of the optimization process.

The rest of this paper is organized as follows. In Section 2, this paper introduces some research advances of VPP source-network-load-storage collaboration. In Section 3, an operation framework for VPP source-network-load-storage based on cloud-edge-end technology is outlined. In Section 4, the VPP source-network-load-storage scheduling model is established. In Section 5, the proposed VPP source-network-load-storage is subjected to simulation analysis. Finally, several highlights are concluded of this paper in Section 6.

## 2. Literature review

Until now, extensive researches on VPP are mainly limited to parts of source-network-load-storage, including source-storage, source-load-storage, source-network, etc. In the VPP source-network aspect, studies mainly focus on the coordination between distributed energy sources and the power network to enhance system stability and flexibility. In [4], advanced control technologies are employed to optimize the interaction between distributed generation units and the grid, improving voltage regulation, frequency support, and fault management capabilities. The complementary characteristics of various RESs are highlighted in [5], since their diverse generation profiles can compensate for each other's intermittency. In [6–7], an enhanced real-time economic scheduling algorithm is developed to improve the accuracy of output prediction for distributed energy resources. In [8], a resilient restoration method is proposed for VPP power-traffic network, which co-optimizes the deployment of mobile power sources and network reconfiguration. In [9], a multi-energy coordination framework is proposed to enhance the VPP resilience against large-scale disturbances, where consumer-side biogas-solar-wind multi-energy VPPs are topologized by network reconfiguration. In [10], the integration of multi-source coordinated scheduling within VPPs is analyzed, with particular emphasis on its performance in ancillary service markets.

In the VPP source-load-storage and source-network-load aspects, demand response and load shedding strategies [11] are extensively utilized to enhance system flexibility by dynamically adjusting energy consumption behaviors and patterns. In [12], a multi-objective source-network-load coordinated model is proposed and solved an improved memetic algorithm, considering the adjustment of load demand curves. To exploit the inherent similarities between RESs and load profiles, a two-layer VPP optimization framework is proposed in [13] to jointly coordinate ESS and industrial loads. In [14], an economic scheduling optimization model is developed, explicitly considering the effects of temperature variations on PV generation performance. In [15], a SOC-based optimization approach is introduced to minimize overall economic costs, system losses, and line voltage deviations through a penalty function formulation. To investigate the beneficial effect of bi-directional electric-hydrogen conversion on market trading, an optimal scheduling method is proposed in [16] for VPP with reversible solid oxide cells in the electricity market. Given the escalating complexity of dispatchable entities within VPP operations, significant attention has been directed toward enhancing dispatch efficiency [17]. In [18], an economic VPP scheduling model considering the load demand under the mechanism of a peak–valley tariff is proposed. To address the uncertainties associated with renewable energy integration, a novel intelligent methodology is proposed in [19] based on fuzzy cloud theory and honeybee mating algorithm.

In the source-network-load-storage aspect, the maximum utilization of energy resources is achieved through multiple forms of interaction between power sources, grids, loads and energy storage. In [20], a stochastic economical optimization strategy is proposed for VPPs, addressing scenarios both with and without demand response programs. Based on model predictive control, a multi-stage optimal source-network-load-storage scheduling model is proposed in [21] to ensure the

safe and economic operation. In [22], a source-network-load-storage optimization model is introduced, where a point-to-point vehicle charging strategy is implemented. In [23], an integrated source-network-load-storage monitoring platform based on the microgrid in State Grid Shanghai Electrical Power Research Institute and blockchain technology. In [24], a novel multi-stage time scale economic dispatch scheme is proposed for VPPs, including a fuzzy optimized day ahead scheduling scheme, an intra-day scheduling scheme combined with Deep Q Network, and an adaptive optimized real-time scheduling scheme. In [25], a bi-layer low-carbon scheduling model is proposed based on node carbon emission intensity. The integration of cloud-edge-end technologies in VPPs is becoming increasingly necessary due to the growing complexity of managing distributed energy resources and ensuring grid stability. VPPs rely on real-time data processing and coordination between various sources of generation, storage, and demand, which can be geographically dispersed and highly dynamic. Together, cloud-edge-end frameworks enhance the flexibility, resilience, and efficiency of VPPs, enabling them to better respond to fluctuations in energy supply and demand, especially in the context of a rapidly evolving electricity-carbon market. However, multi-timescale cloud-edge-end based the VPP source-network-load-storage in the context of electricity-carbon market has not been explored.

Real-time pricing (RTP) and time of use (TOU) are two typical pricing mechanisms to facilitate the VPP source-network-load-storage interactions. While RTP reflects the actual cost of electricity based on real-time supply and demand conditions in the market, TOU is set based on predefined time periods, typically distinguishing between peak, off-peak, and mid-peak hours. Though RTP and TOU pricing mechanisms drive more intelligent interactions between VPP source-network-load-storage, the VPP supply–demand behavior is generally ignored. While the end-users can receive financial benefits from the pricing mechanisms, overall system load profile cannot be flattened. Table 1 summarizes the differences of typical works.

## 3. Problem formulation

Aimed to reduce operating and carbon trading costs, this paper offers guidance on constructing a flexible, multi-party interactive power system. However, with the increasing number of distributed energy sources, the coordinated scheduling of VPP source-network-load-storage is a challenging problem: 1) Distributed resources are characterized by small capacity, large volume, and wide dispersion. To achieve the integration of multiple resources and optimize their output, it is necessary to coordinate the source-network-load-storage to cope with the RES volatility and intermittency. 2) In order to adapt to different time scales (such as long-term day-ahead dispatch and short-term real-time scheduling) and spatial scales (such as local and global coordination), the source-network-load-storage optimization is also a multi-faceted consideration. 3) In order to process massive real-time data and complex calculations, the cloud-edge-end collaborative architecture is applied to VPP. Therefore, the VPP source-network -load-storage optimization is a sophisticated problem, involving

Table 1. Differences with proposed approach.

| References | Source-network | Source-load-storage | Source-network-load | Source-network-load-storage | Pricing | Cloud-edge-end |
|---|---|---|---|---|---|---|
| [4,5,9] | √ | | | | TOU | |
| [6,7,8] | √ | | | | RTP | |
| [11] | | | √ | | TOU | √ |
| [12,14] | | | √ | | TOU | |
| [13,15,18] | | √ | | | TOU | |
| [10,16,19] | | √ | | | RTP | |
| [17] | | | √ | | RTP | |
| [20] | | | | √ | RTP | |
| [21–25] | | | | √ | TOU | |
| Proposed | | | | √ | interactive RTP | √ |

the multiple resources, coordination at multiple time-space scales, real-time data processing, flexible user participation, and the satisfaction of complex system constraints.

## Nomenclature

| Indices and sets | |
|---|---|
| $t$ | Index for time slots |
| $i, j$ | Index for node and branch |
| $\Omega_{wt}, \Omega_{pv}, \Omega_{ess}$ | Set of WT, PV, ESS nodes |
| $\Omega_{sub}$ | Set of substation nodes |
| $\Omega_L, \Omega_n$ | Set of lines and load nodes |
| $\Omega_{ev}$ | Set of EV charging station nodes |
| **Parameters** | |
| $P_{i,t}^{WT}, P_{i,t}^{PV}$ | WT and PV outputs |
| $WT_i, PV_i, ES_i$ | capacities of WT, PV and ESS |
| $\lambda^{WT}, \lambda^{PV}$ | Maximum allowable WT and PV curtailments |
| $K^{sub}$ | Maximum load ratio of the substation |
| $\xi_i$ | Maximum load shedding ratio |
| $P_{i,t}^{LOAD}, Q_{i,t}^{LOAD}$ | Active and reactive load |
| $k_{ch}, k_{dis}, \eta_{ch}, \eta_{dis}, \eta_{ess}$ | Charging-discharging depths, charging-discharging efficiency, stations efficiency |
| $U_{min}, U_{max}, V_0$ | Minimum, maximum, and standard voltage |
| $r_{ij}, x_{ij}, S_{ij}$ | Resistance, reactance and capacity of lines $ij$ |
| $f^{wt}, f^{pv}$ | Penalties for WT and PV curtailment |
| $f_{es}^{op}$ | Battery degradation cost |
| $f_i^n, F_i^{cut}$ | Load shedding penalties |
| $SOC_{min}^{EV}, SOC_{max}^{EV}$ | SOC limits of EVs |
| $P_{max}^{ev}, P_{min}^{ESS,ch}, P_{min}^{ESS,dis}$ $P_{max}^{ESS,ch}, P_{max}^{ESS,dis}$ | Maximum and Minimum charging and discharging power |
| $\lambda_1, \lambda_2, L_1, L_2$ | Real-time pricing coefficient |
| $\mu_{M,buy}, \mu_{M,sell}$ | Price of buying and selling electricity |
| $\sigma_{M,buy}, \sigma_{M,sell}, \sigma_B$ | Coefficients of utility grid, BES, CHP, P2G, and gas tank |
| $P_{CO2}$ | Daily carbon trading prices |
| **Variables** | |
| $P_{i,t}^{wt,use}, P_{i,t}^{wtgu}, P_{i,t}^{pv,use}, P_{i,t}^{pvgu}$ | Consumed and curtailed power of WT and PV |
| $y_{i,j}$ | Virtual flow from node $i$ to node $j$ |
| $P_{i,t,s}^{cut}$ | Curtailed power |
| $P_{i,t}^{Fcut}, Q_{i,t}^{Fcut}$ | Curtailed active and reactive power |
| $P_{i,t}^{ev}$ | EV charging station load |
| $P_{i,t}^{dis}, P_{i,t}^{ch}, \eta_{i,t}^{ch}, \eta_{i,t}^{dis}$ | ESS discharging and charging power and indicators |
| $E_{i,t}^{CO2}$ | Carbon footprint |
| $S_{i,t}^{soc}$ | ESS capacity |
| $P_{i,t}^{in}, Q_{i,t}^{in}, P_{ij,t}^{in}, Q_{ij,t}^{in}$ | Active and reactive power injection |
| $U_{i,t}$ | Voltage at node $i$ |
| $SOC_{i,t,n}^{ev}, SOC_{i,t,n}^{ESS}$ | SOC of Storage and charging stations and EVs |
| $P_{i,t,n}^{ev}, P_{i,t,n}^{ESS,ch}, P_{i,t,n}^{ESS,dis}$ | Charging and discharging power |

| | |
|---|---|
| $U_{i,t,n}^{ch}, U_{i,t,n}^{dis},$ | Binary variables for storage and charging stations |
| $P_{buy}(t), P_{SELL}(t)$ | Power bought and sold |
| $L(t)$ | Net load |
| $Pr(L(t)), TRTP(t)$ | Transactive real-time pricing |
| $P_{ESS}^{disc}, P_{ESS}^{chat}$ | The sum of charging and discharging |
| $R_{M,buy}(t), R_B(t)$ | Penalty costs of utility grid, BES |

## 3.1. Cloud-edge-end collaboration

Given the inherent attributes of distributed resources (namely their small capacities, large quantities, and wide dispersal), a collaborative cloud-edge-end architecture has been devised. As shown in Fig 1, cloud-edge-end collaboration involves the integration of three key components: centralized cloud computing, decentralized edge computing, and end-user devices (or endpoints), which is designed to be efficient and scalable. It is capable of managing data, processing tasks, and providing real-time services [26]. By distributing tasks across each layer, this cloud-edge-end approach optimizes computing efficiency, reduces latency, enhances data processing speed, and lowers bandwidth costs.

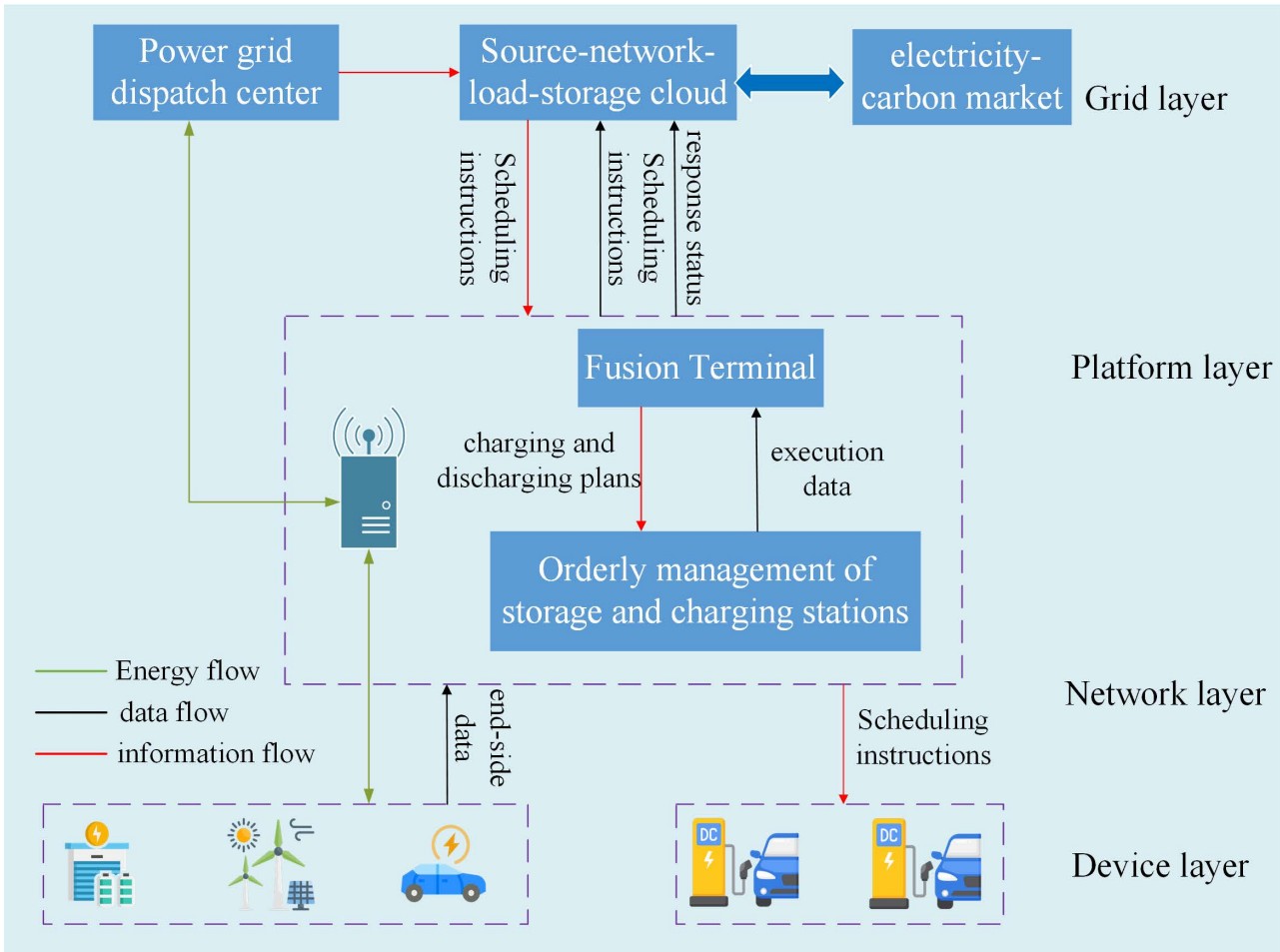

**Fig 1. Cloud-edge-end-based economical management of VPP source-network-load-storage in electricity-carbon market.**

1) Cloud computing: The cloud layer serves as the centralized infrastructure where large-scale data processing, storage, and analytics occur. It provides the computational power to handle complex tasks, manage vast amounts of data, and support advanced algorithms that require significant resources. The cloud is also responsible for orchestrating communication and data flow across the edge and end devices, ensuring synchronization and consistency.

2) Edge computing: The edge layer facilitates data analysis at the network's edge, typically at intermediate nodes (like local servers or gateways) that are geographically closer to end devices or sensors. By processing data locally, edge computing reduces latency, minimizes bandwidth usage, and provides faster response times for time-sensitive applications. Edge devices often perform tasks like data filtering, aggregation, and preprocessing before sending only relevant information to the cloud for further analysis.

3) End devices/end-users: The end layer represents the devices or users interacting directly with the system. These can include smartphones, IoT devices, sensors, or any other consumer-facing or industrial devices. These devices collect data from the physical world and enable real-time decision-making, through collaborating with cloud and edge computing.

Here, cloud-edge-end-based economical management of VPP source-network-load-storage is developed in the cloud-edge side and real-time scheduling of EV storage and charging stations is developed in the end-side.

### 3.2. VPP source-network-load-storage coordination

As shown in Fig 2, VPP source-network-load-storage coordination refers to the integrated management of wind-solar renewable energy sources, distribution network, demand response loads, and battery energy storage. The goal of source-network-load-storage coordination is to enable the optimal, reliable, and sustainable operation of modern energy systems by ensuring coordinated decision-making and interaction among diverse energy sources:

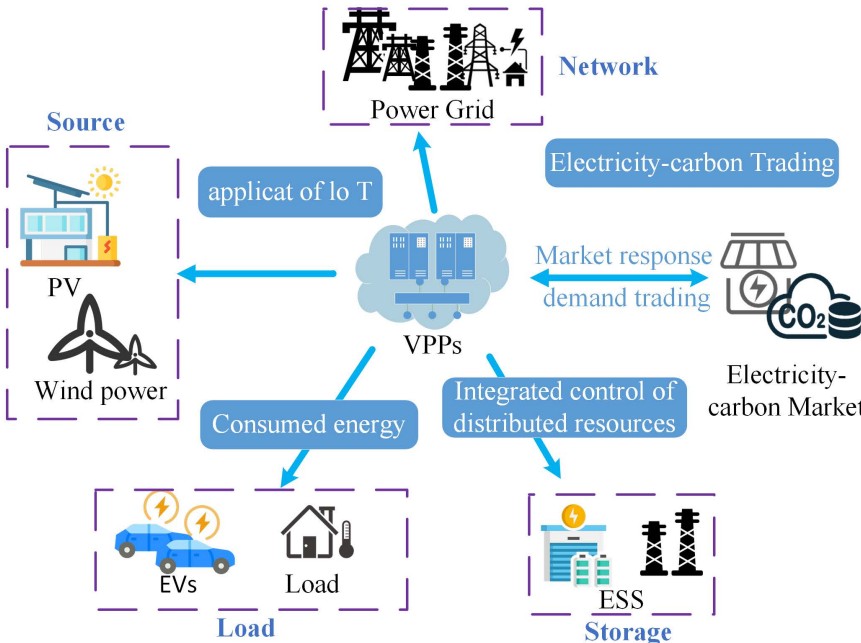

**Fig 2. VPP source-network-load-storage coordination in the context of electricity-carbon market.**

1) Source: Energy generation, particularly from renewable sources, can be unpredictable due to weather conditions. Effective coordination helps align generation with demand, ensuring system optimal resource utilization.

2) Network: The distribution networks are managed to efficiently deliver energy from sources to loads while avoiding congestion, losses, or instability.

3) Load: Managing the demand side involves balancing consumption with available energy supply. This may include using flexible demand (e.g., shifting energy use to off-peak periods) or demand response programs to help smooth out peaks.

4) Storage: When the demand is less than the supply, energy storage systems store excess energy. Conversely, during times of high demand or insufficient renewable generation, the stored energy is released.

### 3.3. EV storage and charging stations with transactive real-time pricing

As a huge distributed energy storage system, EVs are an important and flexible resource for maintaining energy supply-demand balance.

The SOC and charging power of EVs must satisfy the following constraints:

$$\begin{cases} SOC_{\min}^{ev} \leq SOC_{i,t,n}^{ev} \leq SOC_{\max}^{ev} \\ SOC_{i,t,n}^{ev} = SOC_{i,t-\Delta t,n}^{ev} + \eta_{ch}^{ev} P_{i,t-1,n}^{evch} \Delta t / E_{EV,n} \end{cases}$$

(1a)

$$0 \leq P_{i,t,n}^{ev} \leq P_{i,t,n\,\max}^{ev}$$

(1b)

For EV storage and charging stations, the surplus energy is stored when the grid output exceeds the charging demand. Conversely, the storage and charging stations are discharged to supply the needed energy for EV battery.

In order to avoid energy storage degradation due to overcharging and over-discharging, the energy storage range must be limited. Energy storage constraints for EV charging stations are:

$$P_{\min}^{ESS,ch} U_{i,t,n}^{ch} \leq P_{i,t,n}^{ESS,ch} \leq P_{\max}^{ESS,ch} U_{i,t,n}^{ch} \quad \forall t \in [1, t_L]$$

(1e)

$$P_{\min}^{ESS,dis} U_{i,t,n}^{dis} \leq P_{i,t,n}^{ESS,dis} \leq P_{\max}^{ESS,dis} U_{i,t,n}^{dis} \quad \forall t \in [1, t_L]$$

(1d)

$$U_{i,t,n}^{ch} + U_{i,t,n}^{dis} \leq 1 \quad \forall t \in [1, t_L]$$

(1e)

$$E_{i,t+1,n}^{s} = E_{i,t,n}^{s} + \eta_{ESS} P_{i,t+1,n}^{ESS,ch} \Delta t - \frac{1}{\eta_{ESS}} P_{i,t+1,n}^{ESS,dis} \Delta t \quad \forall t \in [1, t_L]$$

(1f)

Despite exhibiting promising performance in EV charging management, both RTP and TOU pricing schemes overlook the supply-demand behavior, failing to effectively alleviate overall system supply-demand imbalances [27]. In order to solve this problem, an interactive real-time pricing mechanism is proposed to achieve the effective charging-discharging management and reliable supply-demand balance.

$$TRTP(t) = c_1 L^2(t) + c_2 L(t) + c_3 \tag{2a}$$

where $L(t)$ denote the net load:

$$L(t) = P_L(t) - P_{buy}(t) + P_{sell}(t) - P_{PV}(t) - P_{WT}(t) \tag{2b}$$

The pricing constraints are:

$$pr(L(t)) = \begin{cases} \mu_{sell} & L(t) < 0 \\ TRTP(t) & 0 < L(t) < L_1 \\ \lambda_1 * TRTP(t) & L_1 < L(t) < L_2 \\ \lambda_2 * TRTP(t) & L(t) > L_2 \end{cases} \tag{2c}$$

When $L(t) \leq 0$, it indicates an abundance of RESs, allowing for the electricity sale at a fixed retail price $\mu_{sell}$. Conversely, when $L(t) > 0$, $pr(L(t))$ is the price of buying electricity, which varies linearly with the net load.

## 4. Cloud-edge-end-based economical management of VPP source-network-load-storage in the context of electricity-carbon market

In the context of electricity-carbon market, the two-layer optimization framework aims to minimize overall costs and reduce renewable energy uncertainties by multi-time scale economical management:

1) In the upper layer, VPP source-network-load-storage is coordinated over a long-term horizon, with supply-demand uncertainties captured by Monte Carlo simulation method. VPP operational cost is optimized to provide scheduling guidelines for the second layer.

2) In the lower layer, EV storage and charging stations are managed to mitigate renewable energy fluctuations through a transactive real-time pricing mechanism. Based on the economic dispatch references from the previous layer, an MPC-based real-time scheduling is performed for short time scale. The optimization results are then fed back to VPP to adjust the relevant parameters. This layer aims to address short-term fluctuations and minimize power regulation costs.

### 4.1. Source-network-load-storage constraints

(1)     Source-side constraints

In the source side, wind power and photovoltaic are integrated in VPP.

$$0 \leq P_{i,t}^{wt,use} \leq WT_i \cdot P_{i,t}^{WT} \quad i \in \Omega_{wt} \tag{3a}$$

$$0 \leq P_{i,t}^{pv,use} \leq PV_i \cdot P_{i,t}^{PV} \quad i \in \Omega_{pv} \tag{3b}$$

$$P_{i,t}^{wt,use} + P_{i,t}^{wtgu} = WT_i \cdot P_{i,t}^{WT} \quad i \in \Omega_{wt} \tag{3c}$$

$$P_{i,t}^{pv,use} + P_{i,t}^{pvgu} = PV_i P_{i,t}^{PV} \quad i \in \Omega_{pv} \tag{3d}$$

$$0 \leq \sum_{i=1}^{n} \sum_{t=1}^{T} P_{i,t}^{wtgu} \leq \lambda^{WT} \sum_{i=1}^{n} \sum_{t=1}^{T} WT_i \cdot P_{i,t}^{WT} \quad i \in \Omega_{wt} \tag{3e}$$

$$0 \leq \sum_{i=1}^{n} \sum_{t=1}^{T} P_{i,t}^{pvgu} \leq \lambda^{PV} \sum_{i=1}^{n} \sum_{t=1}^{T} PV_i \cdot P_{i,t}^{PV} \quad i \in \Omega_{pv} \tag{3f}$$

Constraints (3a)–(3d) are the power balance constraints for wind power and photovoltaic. Constraints (3e)–(3f) are the maximum wind and solar curtailment. In the formula, $\lambda_{PV}$ and $\lambda_{WT}$ are the maximum allowable wind and solar curtailment rates.

(2)    Network-side constraints

When power is transmitted in the distribution network, a series of related flow constraints must be strictly followed to ensure the stable system operation. This paper adopts the linearized DistFlow flow model to reduce the complexity of model calculation, by ignoring the non-convex terms of network loss. Existing literatures have confirmed that the impact of network loss on the final result is relatively small. The network-side constraints are:

$$P_{i,t}^{in} + \sum_{(i,j) \in \Omega_L} P_{ji,t} - P_{i,t}^{LOAD} = \sum_{(i,j) \in \Omega_L} P_{ij,t} \quad i \in \Omega_n \tag{4a}$$

$$Q_{i,t}^{in} + \sum_{(i,j) \in \Omega_L} Q_{ji,t} - Q_{i,t}^{LOAD} = \sum_{(i,j) \in \Omega_L} Q_{ij,t} \quad i \in \Omega_n \tag{4b}$$

$$P_{i,t}^{in} = P_{i,t}^{wt,use} + P_{i,t}^{pv,use} + P_{i,t}^{sub} + P_{i,t}^{cut} + P_{i,t}^{dis} - P_{i,t}^{ch} \quad i \in \Omega_n \tag{4c}$$

$$Q_{i,t}^{in} = Q_{i,t}^{svg} + n_{i,t} \cdot Q_{i,t}^{Fcut} \quad i \in \Omega_n \tag{4d}$$

$$U_{i,t} - U_{j,t} \leq M(1 - x_{ij}) + (r_{ij}^L P_{ij,t} + x_{ij}^L Q_{ij,t})/V_0 \quad (i,j) \in \Omega_L \tag{4e}$$

$$U_{i,t} - U_{j,t} \geq M(1 - x_{ij}) + (r_{ij} P_{ij,t} + x_{ij} Q_{ij,t})/V_0 \quad (i,j) \in \Omega_L \tag{4f}$$

$$U_{min} \leq U_{i,t} \leq U_{max} \quad i \in \Omega_n \tag{4g}$$

$$0 \leq \sqrt{(P_{ij,t})^2 + (Q_{ij,t})^2} \leq x_{ij} S_{ij} \quad (i,j) \in \Omega_L \tag{4h}$$

Constraints (4a)–(4b) delineate the power flow among nodes, whereas (4c)–(4d) specify the active and reactive power injections. Voltage drops between nodes are indicated by (4e)–(4f). Constraints (4g) define the acceptable voltage range, and (4h) outlines the line capacity.

**(3)    Load-side constraints**

Load shedding is a strategic approach to controllably reduce electricity demand when the power supply is insufficient to meet the total consumption.

$$0 \le P_{i,t,s}^{cut} \le \xi_i P_{i,t,s}^{LOAD} \quad i \in \Omega_n \tag{5a}$$

Constraint (5a) specifies maximum allowable load shedding.

**(4)    Storage-side constraints**

Energy storage is capable of storing energy for later use, enabling a more reliable, efficient, and sustainable energy supply. Constraint (6a)–(6f) represent the operation of ESS.

$$0 \le P_{i,t}^{dis} \le \eta_{i,t}^{dis} \cdot k_{dis} \cdot ES_i \quad i \in \Omega_{es} \tag{6a}$$

$$0 \le P_{i,t}^{ch} \le \eta_{i,t}^{ch} \cdot k_{ch} \cdot ES_i \quad i \in \Omega_{es} \tag{6b}$$

$$\eta_{i,t}^{ch} + \eta_{i,t}^{dis} = 1 \quad i \in \Omega_{es} \tag{6c}$$

$$S_{i,t}^{soc} = S_{i,t-1}^{soc} + \eta_{ch} P_{i,t}^{ch} \Delta t - \frac{P_{i,t}^{dis} \Delta t}{\eta_{dis}} \quad i \in \Omega_{es} \tag{6d}$$

$$\mu_{min} \cdot ES_i \le S_{i,t}^{soc} \le \mu_{max} \cdot ES_i \quad i \in \Omega_{es} \tag{6e}$$

$$S_{i,1}^{soc} = S_{i,24}^{soc} \quad i \in \Omega_{es} \tag{6f}$$

Constraints (6a)–(6c) define the charging and discharging upper bounds. Constraints (6d)–(6f) enforce the energy storage capacity limits within its specified range.

## 4.2. System operational constraints

For the upper-layer day-ahead optimization, the following should be satisfied:

$$P_{i,t_u}^{L} = P_{i,t_u}^{buy} - P_{i,t_u}^{sell} + P_{i,t_u}^{B,dis} - P_{i,t_u}^{B,ch} + P_{i,t_u}^{PVT} + P_{i,t_u}^{WT} \tag{7a}$$

Among them, $t_u$ represents the optimization time slot of the upper layer.
   For the real-time scheduling in the lower layer, the following energy balance equation needs to be satisfied:

$$P_{i,t_l}^{L} = P_{i,t_l}^{buy} - P_{i,t_l}^{sell} + P_{i,t_l}^{B,dis} - P_{i,t_l}^{B,ch} + \sum P_{i,t_l}^{ev,dis} - \sum P_{i,t_l}^{ev,ch} + P_{i,t_l}^{e,dis} - P_{i,t_l}^{e,ch} + P_{i,t_l}^{PVT} + P_{i,t_l}^{WT} \tag{7b}$$

Among them, $t_l$ represents the optimization time slot of the lower layer.

Similarly, the related source-network-load-storage constraints in equations (3a)–(6f) should also be considered for the lower layer.

## 4.3. Formulation for upper-layer

In the upper-layer, the optimization goal is to minimize the system operation cost, which is comprised of battery degradation, wind-solar curtailment, and load shedding.

The upper-layer objective function can be formulated as follows:

$$F_u : \min \left( \sum_{t_l \in \{1,...,T_u\}} (C_B(t_u) + C_G(t_u) + C_n(t_u)) + C_{CO2} \right) \tag{8a}$$

$$C_B(t_u) = \sum_{i \in \Omega_{es}} f_{es}^{op}(P_{i,t}^{ch} + P_{i,t}^{dis}) \tag{8b}$$

$$C_G(t_u) = \sum_{i \in \Omega_{wt}} f^{wt} P_{i,t}^{wtgu} + \sum_{i \in \Omega_{pv}} f^{pv} P_{i,t}^{pvgu} \tag{8c}$$

$$C_n(t_u) = \sum_{i \in \Omega_n} f^n P_{i,t}^{cut} \tag{8d}$$

$$C_{CO2} = \sum_{i \in \Omega_n} p_{CO2} E_{i,t}^{CO2} \tag{8e}$$

Constraints (8b) are battery degradation cost; Constraints (8c) are wind and solar power curtailment cost; Constraints (8d) are load shedding cost. (8e) is carbon trading costs.

## 4.4. Formulation for lower layer

After obtaining the scheduling reference from the upper layer, the lower layer aims to obtain the minimum deviation between the actual output and the scheduling plan. EV storage and charging stations play a crucial role in maintaining supply-demand balance, which can effectively address the RES uncertainty.

Based on the transactional real-time electricity price, the economic cost of the storage and charging station $C_{EV}(t)$ is the discharge and charging cost:

$$C_{EV}(t) = pr(L(t))P_{ESS}^{disc}(t) - \mu_{sell}P_{ESS}^{char}(t) \tag{9a}$$

The deviation from the first layer scheduling reference can be expressed as quadratic terms:

$$R_{M,buy}(t_l) = (P_{M,buy}(t_u) - P_{M,buy}(t_l))^2$$
$$R_{M,sell}(t_l) = (P_{M,sell}(t_u) - P_{M,sell}(t_l))^2$$
$$R_B(t_l) = (SOC_{BES}(t_u) - SOC_{BES}(t_l))^2 \tag{9b}$$

The objective function of the second layer is formulated by incorporating weighted quadratic penalty terms:

$$F_I : min \left( \begin{array}{c} \sum\limits_{t_l \in \{1,...,T_l\}} \sum C_{EV}(t_l) \\ + \sum\limits_{t_l \in \{1,...,T_l\}} \left( \begin{array}{c} \sigma_{M,buy}R_{M,buy}(t_l) + \sigma_{M,sell}R_{M,sell}(t_l) \\ +\sigma_B R_B(t_l) \end{array} \right) \end{array} \right) \tag{9c}$$

The weighting factors can be varied according to importance, but their sum is limited to 1.

### 4.5. MILP reformulation and solving

After applying piecewise linearity to equations (2a) and (2c), the remaining nonlinearities are bilinear terms in equation (9a). These bilinear terms are addressed using the McCormick envelope method [28]. By introducing new variables to replace the bilinear terms, the problem is linearized by establishing corresponding constraints for these auxiliary variables. In this paper, let $A_{i,k,j} = pr(L(t)) \cdot P_{ESS}^{disc}(t)$ and formulate constraints:

$$P_{n,dis}pr(L(t))_{min} \leq A_{i,k,j} \leq P_{ch,dis,max}pr(L(t)) + P_{ESS}^{disc}(t)pr(L(t))_{min} - P_{ch,dis,max}pr(L(t))_{min} \tag{9d}$$

$$P_{ch,dis,max}pr(L(t)) + P_{ESS}^{disc}(t)pr(L(t))_{min} - P_{ch,dis,max}pr(L(t))_{min} \leq A_{i,k,j} \leq P_{n,dis}pr(L(t))_{max} \tag{9f}$$

Fig 3 is a two-layer optimization flowchart of the proposed VPP source-network-load-storage. The entire two-layer optimization model in this study is formulated as a MINLP problem, which cannot be easily solved using off-the-shelf solvers directly. In such cases, the original MINLP model is first linearized using the McCormick envelope method [28], and then can be equivalent to an LP model by removing binary variables [29]. This results in a convex optimization problem with a unique global optimum, ensuring efficient and reliable scheduling.

The uncertainty associated with renewable energy generation is modeled using a Monte Carlo scenario-based approach [27]. A set of representative scenarios is generated to reflect the stochastic nature of renewable resources, considering historical data and probabilistic distributions. These scenarios are incorporated into the upper-layer optimization to ensure robust scheduling decisions under uncertain conditions. Additionally, the end-side control layer employs MPC, which further mitigates real-time uncertainty by continuously updating the control strategy based on the latest system information.

## 5. Case study

### 5.1. Base data

This paper uses a 34-node system to simulate the proposed VPP source-network-load-storage in the context of electricity-carbon market. Wind farms are built at nodes 13 and 16, with installed capacities of 15MW and 6MW respectively. Photovoltaic generations are built at nodes 5 and 10, with installed capacities of 14MW and 15MW respectively. Energy storage stations are built at nodes 17 and 31, with capacities of 4MW and 5MW respectively. Parameters of VPP source-network-load-storage are acquired from [28,29], which are listed in Table 2. Fig 4 gives the grid electricity price for market. The two-layer optimization scheme covers a 24-hour period, with time slots set to 1 hour and 15 minutes for the upper and lower layers, respectively. In Section 4.2, a comparative analysis is conducted between the proposed two-layer optimal scheduling strategy and a single-layer scheme [4] to validate the superior performance of the hierarchical coordination approach. Similarly, in Section 4.3, the effectiveness of the interactive real-time pricing mechanism is evaluated through comparisons with TOU [30] and RTP [31] schemes.

### 5.2. Comparative results with existing optimization schemes

The VPP load status and the charging/discharging status of the energy storage system are depicted in Figs 5,6. In the early morning, the load is relatively low but the wind power is relatively strong. During this period, the energy storage

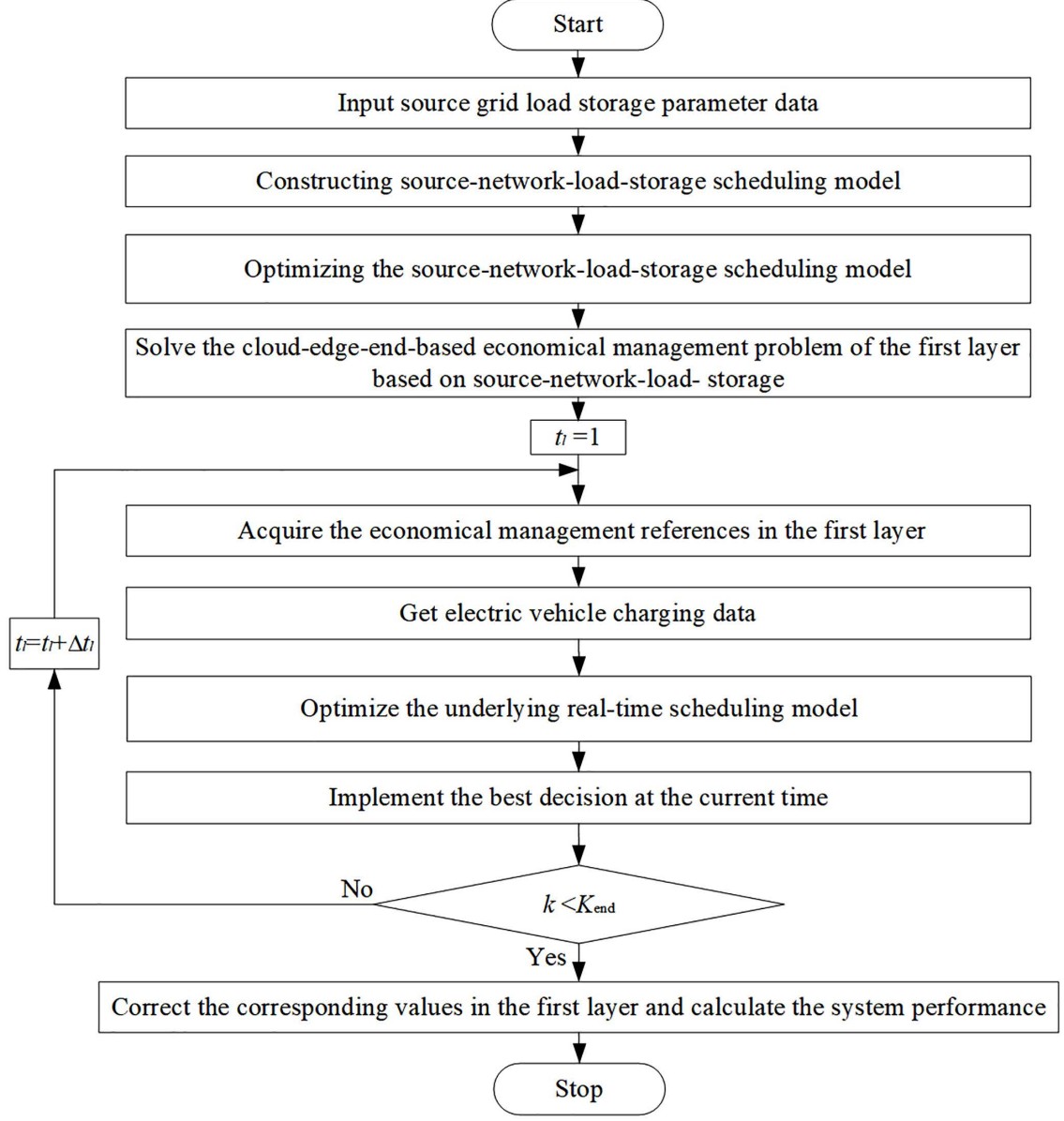

**Fig 3. Cloud-edge-end-based economical management of VPP source-network-load-storage.**

system begins storing excess electricity to supply energy later. Between 17:00 and 20:00, the load increases, while the wind and solar power generation decreases. The energy storage begins to discharge to make up to balance the supply and demand.

To assess the economic benefits of the proposed two-layer VPP source-network-load-storage optimization, a comparative analysis involving both single-layer and two-layer schemes was conducted.

Table 3 presents a comparative analysis of the operating costs for the source-network-load-storage components of a VPP under two different optimization schemes. In the proposed two-layer scheme, the EV storage and charging stations are capable of storing energy during low-cost periods and subsequently discharging it for sale during peak demand. This

**Table 2. Simulation parameters.**

| Source-network-load-storage | $k_{dis}$ =0.2 | | $k_{ch}$ =0.2 |
|---|---|---|---|
| | $\eta_{ch}$ =0.9 | | $\eta_{dis}$ =0.9 |
| | $U_{min}$ = 14.25kV | | $U_{max}$ = 15.57kV |
| Storage and Charging Station | $P_{ch,dc,max}$ = 20 kw | | $P_{dis,dc,max}$ = 10 kw |
| | $Q_e$ = 50 kwh | | $w_{100}$ = 30 kwh |
| | $SOC_{EV,min}$ = 0.1 | | $SOC_{EV,max}$ = 1 |
| | $SOC_e$ = 0.8 | | |
| Objective Function | $\sigma_{M,buy}$ = 0.35 \$/$kw^2$ | | $\sigma_{M,sell}$ = 0.1 \$/$kw^2$ |
| | $\sigma_B$ = 0.2 \$/unit | | |

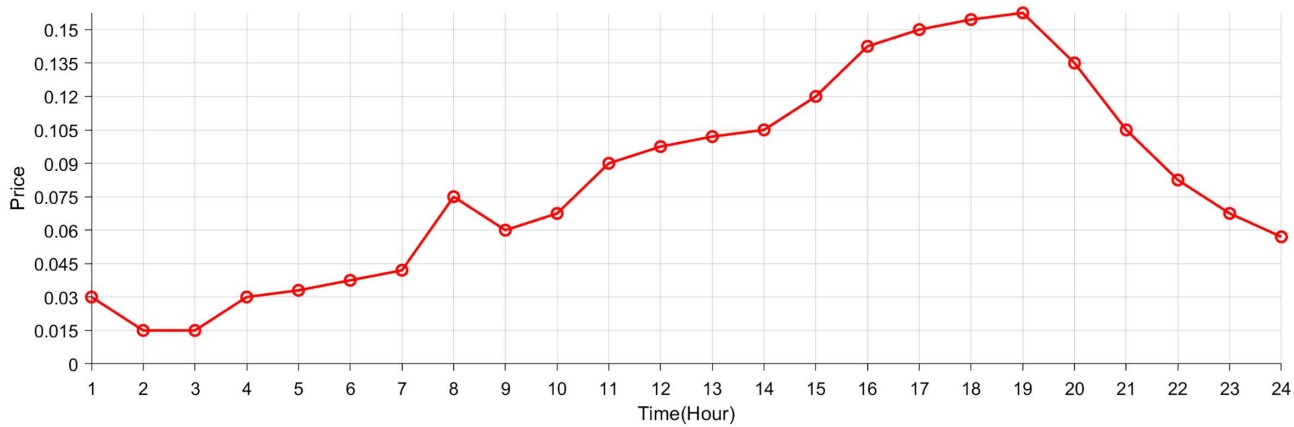

**Fig 4. Grid electricity price.**

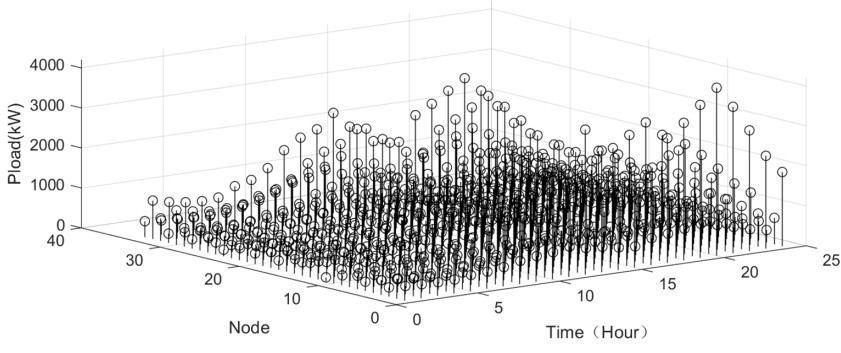

**Fig 5. Load status.**

reduces the frequency of charging and discharging cycles in the energy storage system, thereby improving economic returns for the VPP.

Moreover, the two-layer optimization strategy enhances the utilization efficiency of renewable energy sources, such as photovoltaic and wind power. By minimizing the dependence on electricity procurement from thermal power units, the scheme significantly reduces carbon emissions associated with fossil-fuel-based generation during electricity market

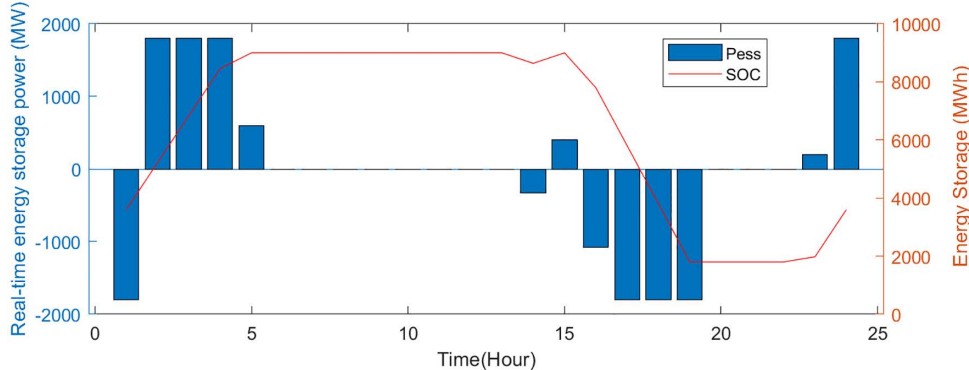

**Fig 6. Energy storage system charging and discharging status.**

**Table 3. Comparison of optimization results between single layer and two-layer.**

| Scheme | One-layer | Two-layer |
|---|---|---|
| Buying and selling electricity (kWh) | 259158.14 | 258552.8926 |
| Storage and charging stations cost ($) | \ | 30.88 |
| Carbon trading costs ($) | 2237.31 | 2190.36 |
| Total cost ($) | 5011.44 | 4896.29 |

transactions, thus lowering the overall carbon trading costs. Although the two-layer optimization strategy incurs an additional scheduling expense of $30.88 compared to the single-layer approach, the overall operational cost remains lower than that of the single-layer scheme. Compared to the single-layer optimization strategy, the two-layer approach proposed in this paper achieves a 2.35% reduction in total operating costs and a 2.14% decrease in carbon trading expenses. However, due to capacity and power output constraints, the contribution of EV storage and charging stations to the overall power supply remains limited and has no pronounced effect.

Figs 7,8 show the buying-selling electricity, energy storage system and station SOC under the two-layer optimization scheme. Compared to the single-layer scheme, the proposed two-layer optimization yields similar optimization results in certain aspects. However, the two-layer scheme distinguishes itself by efficiently managing multiple energy flows, enabling better adaptation to RES fluctuations. During off-peak hours (such as early mornings) when electricity prices are low, consumers are inclined to buy electricity from the electricity-carbon market to satisfy their energy needs. In the evening hours, VPP purchases electricity from the electricity-carbon market due to increased load. In the noon and afternoon hours, VPP sells electricity to the market to generate revenue, since photovoltaic and wind turbines provide a large amount of electricity.

Figs 9–11 show the storage SOC and carbon trading cost results. As the EV storage and charging station is involved in the second-layer optimization, its energy storage capabilities help alleviate the supply-demand imbalance, resulting in a less pronounced variation in the SOC. Since EV storage and charging stations are used to adjust power fluctuations, the SOC curves of the energy storage system are same in two schemes. Flexible energy supply can be combined with real-time tariffs to effectively utilize clean energy (photovoltaic and wind turbine) generation, reducing system operating costs and carbon emissions.

### 5.3. Comparative results with existing pricing schemes

The SOC curves of the EV storage and charging station are shown in Fig 12. In the proposed scheme, the EV storage and charging station charges during the early morning hours and discharges during the low-price periods of 9–10, 12–14,

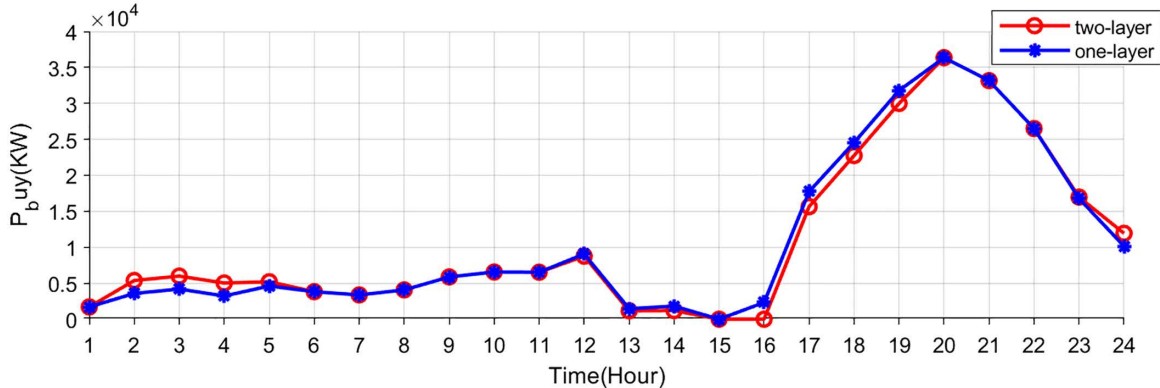

**Fig 7. Electricity bought of electricity-carbon market.**

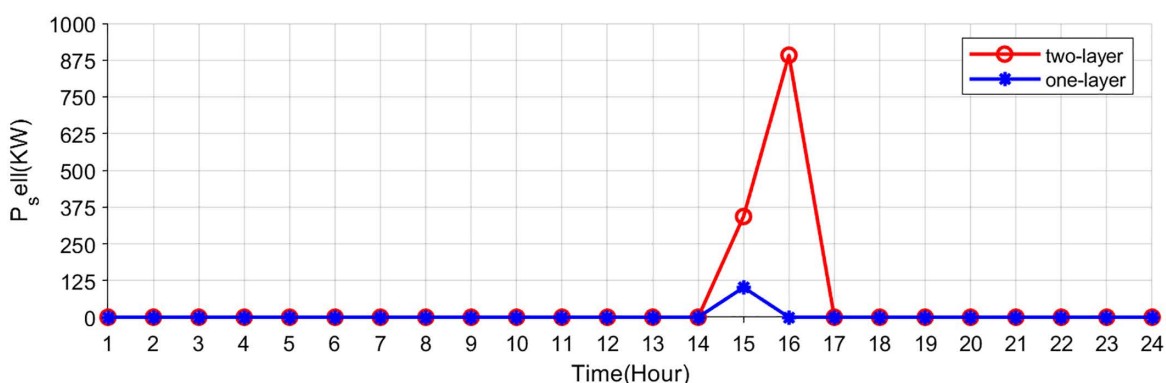

**Fig 8. Electricity sold of electricity-carbon market.**

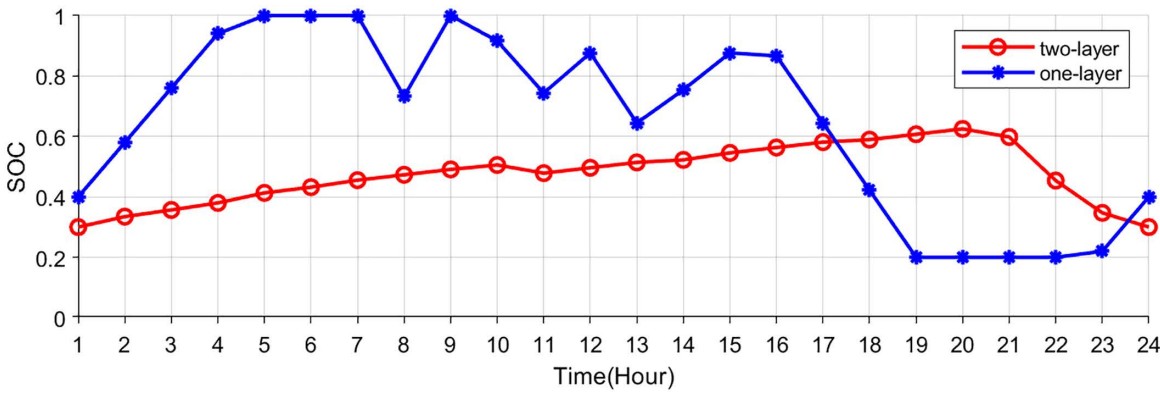

**Fig 9. Total SOC of EV charging station.**

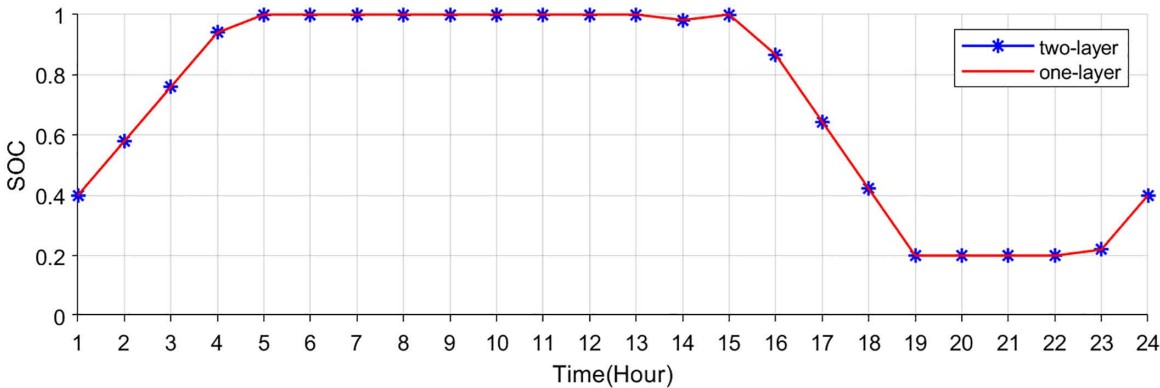

**Fig 10. SOC of energy storage system.**

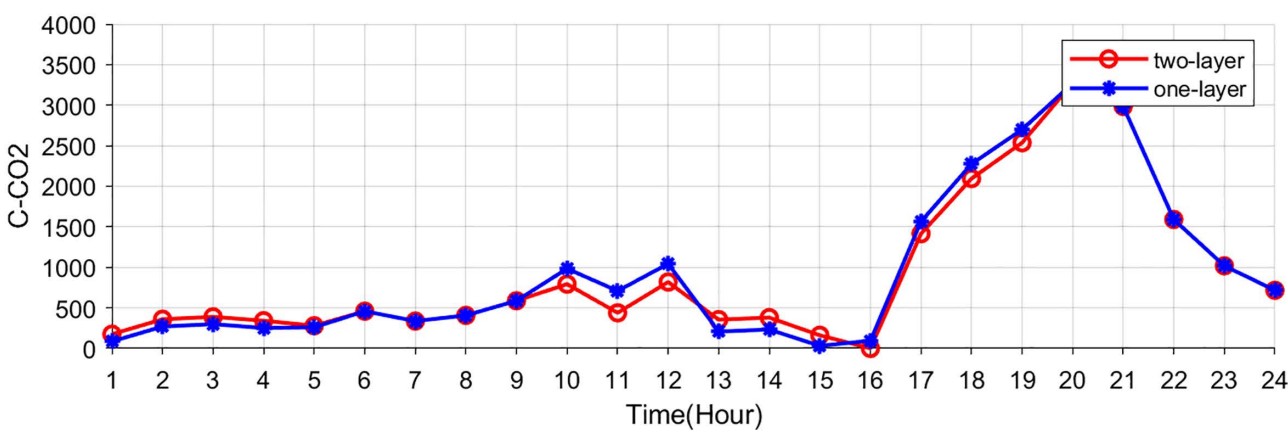

**Fig 11. The cost of carbon trading.**

and 18–19 hours. Conversely, it discharges during the higher price periods of 11–12, 14–15, and 20–24 hours, smoothing the net load effectively.

Fig 13 illustrates the net load profile under three different pricing mechanisms. It can be found that net load fluctuations are effectively smoothed as energy storage systems and EV charging stations respond to price signals under all three pricing mechanisms. Among them, the proposed pricing mechanism based on real-time net load characteristics exhibits the most favorable performance. EV storage systems and charging stations are observed to charge during system surplus periods and discharge during high-price periods in response to real-time market signals, thereby improving overall electricity utilization efficiency. Specifically, EV storage and charging stations strategically charge during periods of electricity surplus, typically characterized by lower prices and high renewable generation output. Conversely, during peak demand or when electricity prices surge, these stations discharge stored energy back into the grid, thereby alleviating supply–demand imbalances. This dynamic interaction not only enhances the temporal utilization efficiency of electricity but also contributes to peak shaving, renewable energy accommodation, and overall system flexibility.

The proposed scheme demonstrates practical feasibility by effectively mitigating the mismatch between electricity supply and demand, while simultaneously reducing the charging and discharging costs associated with EV storage and

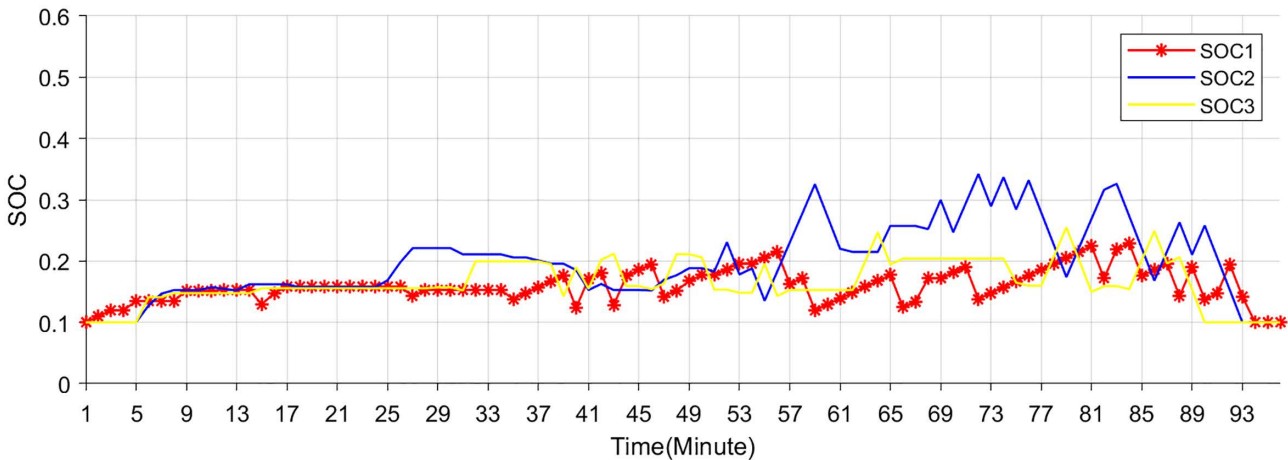

**Fig 12. SOC of Storage and charging stations.**

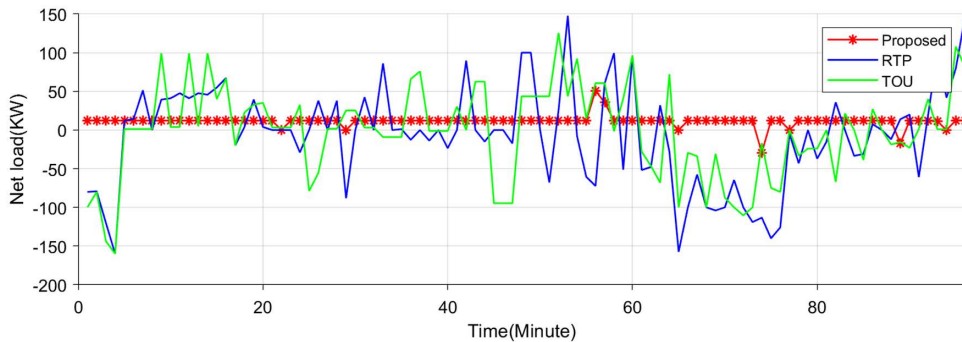

**Fig 13. Net load curves under three pricing schemes.**

charging stations. Compared with the other two electricity pricing strategies, the proposed method results in a reduction of EV-related operational costs by 13.83% and 11.32%, respectively.

## 5.4. Comparative results with storage and charging stations

The motivation for deploying EV storage and charging stations stems from the rapid growth of electric vehicles, the increasing demand for sustainable transportation. EV storage and charging infrastructure play a critical role in promoting decarbonization, enhancing grid resilience, and supporting the transition to a low-carbon energy ecosystem. Here, EV storage and charging stations are used to balance RES fluctuations. The proposed method is compared with a scenario without energy storage stations.

Figs 14,15 shows the energy storage SOC and net load curve. Without the integration of EV storage and charging stations, the net load exhibits sharp peaks and troughs, reflecting high volatility in power demand and a greater reliance on grid flexibility. Such fluctuations can complicate grid scheduling and increase the need for costly ancillary services. With the proposed method, the net load curve becomes significantly smoother. The EV storage and charging stations effectively act as distributed flexibility assets, absorbing excess generation during low-demand periods and supplying energy during peak demand intervals. This not only alleviates grid stress but also enhances system stability and economic

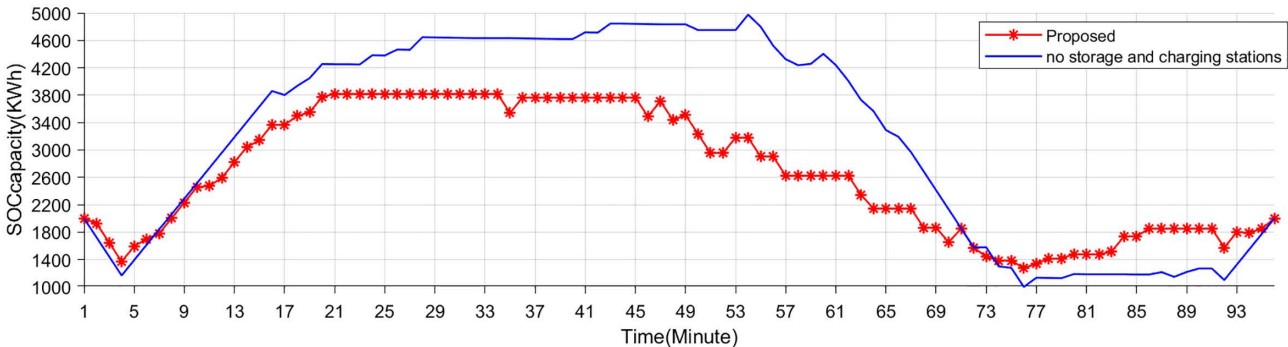

**Fig 14. SOC of the energy storage systems under two scenarios.**

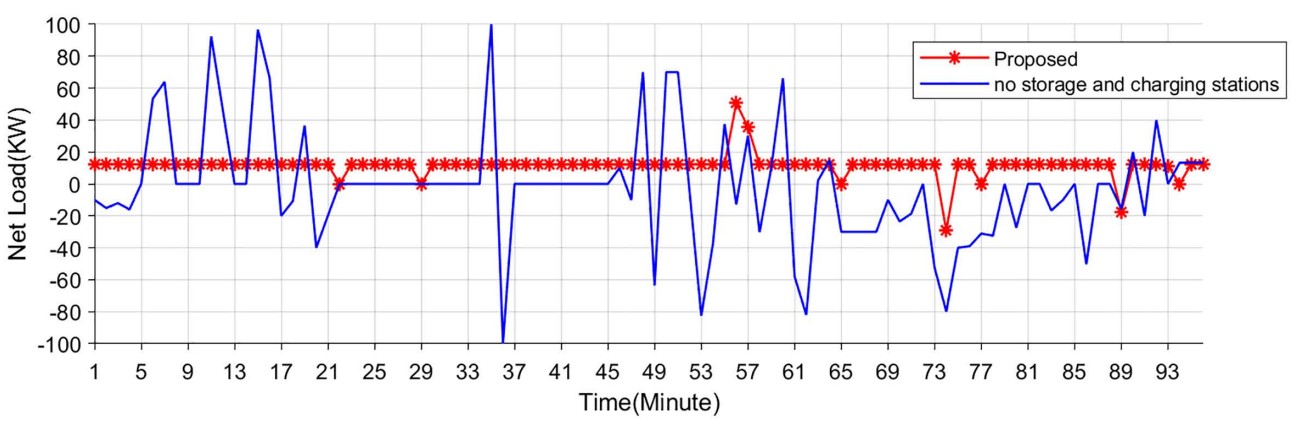

**Fig 15. Net load curves under two scenarios.**

performance. These results collectively highlight the critical role of coordinated EV storage and charging strategies in supporting grid flexibility, reducing operational costs, and promoting the integration of renewable energy sources.

These results collectively highlight the critical role of coordinated EV storage and charging strategies in supporting grid flexibility, reducing operational costs, and promoting the integration of renewable energy sources.

### 5.5. Sensitivity analysis on battery degradation cost coefficient

Battery degradation cost is a key economic parameter that directly affects the decision-making of energy storage. It reflects both the replacement cost of batteries and the marginal cost of usage. To evaluate the sensitivity to this parameter, we conduct a series of simulations by varying the battery degradation cost coefficient from 0.001 to 0.07.

Fig 16 shows the system operating cost with respect to battery degradation cost coefficient. With the increase of battery degradation cost coefficient from 0.001 to 0.06, the total system operating cost gradually rises. The battery is used in a limited manner to circumvent the high battery degradation costs, resulting in increased reliance on grid electricity or alternative resources. With the increase of battery degradation cost reaches 0.06, the degradation cost becomes comparable to or higher than the benefit of battery participation, resulting in avoiding battery operation to minimize system cost. Beyond this threshold, further increases in the degradation cost coefficient do not impact the total cost, as the battery is no longer utilized.

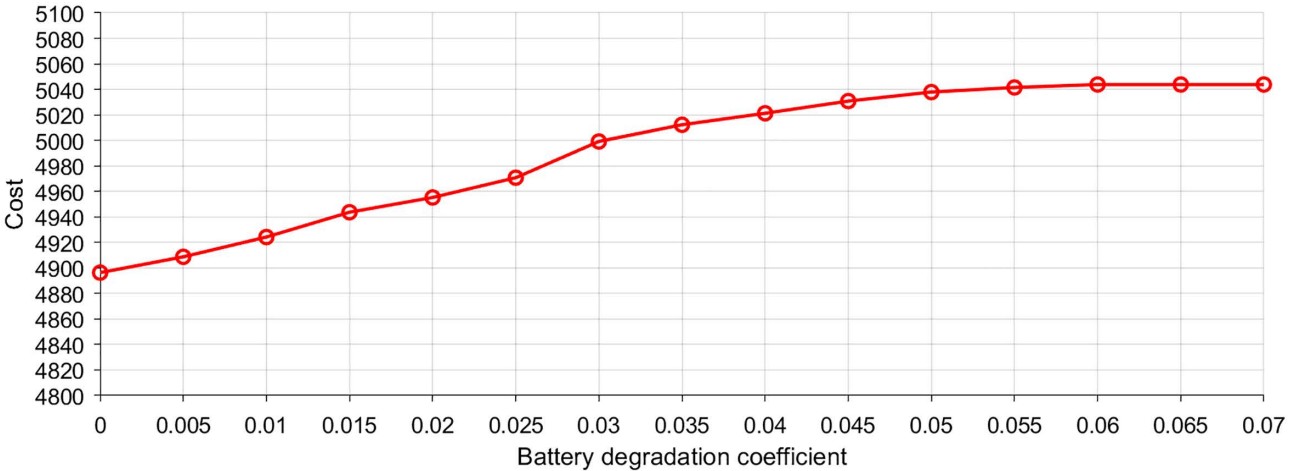

**Fig 16. Sensitivity analysis of system operating cost with respect to battery degradation cost coefficient.**

## 6. Conclusion

This paper presents a cloud–edge–end multi-time scale economic management framework for VPP source-network-load-storage under the electricity–carbon market environment. In the first layer, a day-ahead source-network-load-storage optimization model is formulated to minimize the overall operational cost. In the second layer, an MPC-based real-time strategy is employed for the scheduling of EV storage and charging stations, aiming to mitigate the uncertainty and intermittency associated with RESs. Comparative analysis demonstrates that:

1) By optimizing the interaction among various system components, the coordination of source-network-load-storage enhances the dispatchability of the system across both spatial and temporal domains, thereby reducing at most 13.83% EV cost.

2) The deployment of EV storage and charging stations equipped with real-time pricing mechanisms enables dynamic load balancing and more efficient resource utilization, which leads to a 2.35% reduction in overall operational costs and a 2.14% reduction in carbon trading expenditures.

3) The proposed two-layer optimization approach outperforms other methods in terms of energy savings and net load management, demonstrating significant potential for application in systems with abundant renewable energy sources.

Further research would focus on the application of machine learning techniques to VPP source-network-load-storage management under high-uncertain environment.

## Author contributions

**Conceptualization:** Panhong Zhang.

**Data curation:** Yilu Zheng.

**Formal analysis:** Panhong Zhang.

**Funding acquisition:** Panhong Zhang.

**Investigation:** Yilu Zheng.

**Methodology:** Panhong Zhang, Yilu Zheng.

**Project administration:** Panhong Zhang.

**Resources:** Panhong Zhang.

**Software:** Yilu Zheng.

**Supervision:** Panhong Zhang.

**Validation:** Yilu Zheng.

**Visualization:** Yilu Zheng.

**Writing – original draft:** Panhong Zhang, Yilu Zheng.

**Writing – review & editing:** Panhong Zhang, Yilu Zheng.

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
