## [Decision Letter · Decision Letter 0]

31 Mar 2025

Dear Dr. Zhang,

Thank you for submitting your manuscript to PLOS ONE. After careful consideration, we feel that it has merit but does not fully meet PLOS ONE’s publication criteria as it currently stands. Therefore, we invite you to submit a revised version of the manuscript that addresses the points raised during the review process.

We look forward to receiving your revised manuscript.

Kind regards,

Muhammad Zakarya, PhD

Academic Editor

PLOS ONE

Additional Editor Comments:

The introduction section needs considerable effort (concise and brief). The problem being investigated should be described clearly, but before that, the field of research should be made clearer. Furthermore, briefly describe the major contributions in bullet form, just before the organization paragraph.

The introduction should lead the way throughout the paper. In addition, the benefits coming from this paper should be made clearer in the introduction and throughout the paper.

I suggest summarizing the related works into a table with respect to their characteristics. The authors should put their proposal into this table for easy comparison.

This will make it clearer to readers, and they will be able to see what was missing in the literature and how this is addressed in this paper.

Proofread the article to ensure appropriate use of English grammar, tenses, and punctuation. Longer sentences should be broken out into smaller ones. There are also some linguistic issues that should be corrected. The use of article "the" is redundant and somewhere missing.

Reviewers' comments:

Reviewer's Responses to Questions

**Comments to the Author**

1. Is the manuscript technically sound, and do the data support the conclusions?

Reviewer #1: Partly

Reviewer #2: Yes

2. Has the statistical analysis been performed appropriately and rigorously?

Reviewer #1: No

Reviewer #2: Yes

3. Have the authors made all data underlying the findings in their manuscript fully available?

Reviewer #1: Yes

Reviewer #2: Yes

4. Is the manuscript presented in an intelligible fashion and written in standard English?

Reviewer #1: Yes

Reviewer #2: Yes

Reviewer #1: 1- The manuscript contains several typographical as well as grammatical errors. Authors are suggested to get the manuscript proofread by a native English speaker.

2- The abstract only state the experimental or calculated process of this work. The authors should be given specific result to demonstrate the research significance.

3- The introduction can be improved by providing a more critical discussion of recent related literature. Discuss the shortcomings of previous work and the gaps, and how this work intends to fill those gaps. The authors should determine the novelty of the current research in the last paragraph of the introduction part.

4- Ensure all figures are presented in high resolution with clarity.

5- The authors haven't stated the limitations of their study.

6- Report the statistical analysis in a dedicated section and the significance of results.

7- All the findings of the current work need to be compared and discussed with the results of other researchers finding instead of having a general comparison with other researchers' works. The authors should perform a comparison between the forecasting results. In your discussion section, please link your empirical results with a broader and deeper literature review.

8- # Conclusions: This section is too general, please add some information about the value of obtained parameters of tested the series as well as information about the novelty of the proposed technology.

Reviewer #2: This paper proposes a cloud-edge-end-based economical management of virtual power plant source-network-load-storage in the context of electricity-carbon market. The manuscript presents some significant results worthy of publication but should be revised to be more precise and accurate in the wording of the text.

1. Add some of the most important quantitative results to the abstract. Focus on the advantages of the proposed method with respect to the obtained results.

2. Most of the ideas written were already described in many literatures. The Authors tried to compile it but lack of the enhancement of the interrelation analysis between the references. It is advised that the authors give a deeper analysis on how these ideas become more applicative strategies so that they can contribute to the next step of implementation.

3. The literature review could be greatly improved. The authors first need to make comparisons of the references and then draw the motivation of the paper. Neither the comparison of references and this work nor the corresponding conclusion is made in the paper. Thus, it is difficult for me to know the novelty and advantages of this paper over other works. To improve this part, the following relevant recent publications can be discussed: Optimal scheduling of virtual power plants with reversible solid oxide cells in the electricity market; An Innovative Cloud-Fog-Based Smart Grid Scheme for Efficient Resource Utilization; A novel fuzzy cloud stochastic framework for energy management of renewable microgrids based on maximum deployment of electric vehicles.

4. The relevant discussion and content of the figures and tables in the text are not enough.

5. There are several up-to-date approaches for the idea. Authors should look for these approaches, compare the results and prove their idea.

6. Author must justify the choice of solution method. How do authors guarantee optimality of the obtained solutions?

7. What is the main driver of optimizing cost? Electricity price? Please add figure with electricity price data.

8. The authors must explain how their approach can handle the uncertainties.

9. What's the sensitivity of the model with the change of computational parameters?.

**Do you want your identity to be public for this peer review?** For information about this choice, including consent withdrawal, please see our Privacy Policy

Reviewer #1: No

Reviewer #2: No

---

## [Author Response · Author response to Decision Letter 1]

12 May 2025

The Reviewers have meticulously examined our manuscript and presented us with a very constructive set of comments and requests, which we believe have improved the strength of our manuscript and the presentation of our work significantly. In the following pages, we address each and every comment of all the Reviewers. Each comment is marked by the Reviewer’s and comment’s index and includes the Reviewer’s text in the original form and content for convenience of reference. Each section discusses one comment, and refers to the particular point of our revised manuscript that has been amended to accommodate this comment.

Please refer to the attachment for details.

We have endeavoured to take all the remarks and requests of the Reviewers strongly into account, and have addressed all comments analytically and very carefully, no matter how minor or major they were. We hope our revision is now up to the standards of PLOS ONE and if requested, we would be glad to address any further issues in subsequent revisions.

---

## [Decision Letter · Decision Letter 1]

23 Jun 2025

Dear Dr. Zhang,

Thank you for submitting your manuscript to PLOS ONE. After careful consideration, we feel that it has merit but does not fully meet PLOS ONE’s publication criteria as it currently stands. Therefore, we invite you to submit a revised version of the manuscript that addresses the points raised during the review process.

We look forward to receiving your revised manuscript.

Kind regards,

Muhammad Zakarya, PhD

Academic Editor

PLOS ONE

Journal Requirements:

Additional Editor Comments:

1) The introduction section still needs improvements.

Add more details about the problem being invsetigated in the introduction section.

Descuss how this research adreeses the problem (research methodology).

Remove the literature review (Sec. 1.2) and put it in a separate section.

Merge Sec. 1.1. and 1.3. Moreover add the final paragraph that describes the whole organization and flow of the paper at the end.

2) Proofread the article to ensure appropriate use of English grammar, tenses, and punctuation. Longer sentences should be broken out into smaller ones. There are also some linguistic issues that should be corrected. The use of article "the" is redundant and somewhere missing.

3) The notation table can be put in Sec 2 (problem formulation)

4) Improve the organization of the paper.

Reviewers' comments:

Reviewer's Responses to Questions

**Comments to the Author**

Reviewer #1: All comments have been addressed

Reviewer #2: All comments have been addressed

2. Is the manuscript technically sound, and do the data support the conclusions?

Reviewer #1: Yes

Reviewer #2: Yes

3. Has the statistical analysis been performed appropriately and rigorously?

Reviewer #1: N/A

Reviewer #2: Yes

4. Have the authors made all data underlying the findings in their manuscript fully available?

Reviewer #1: Yes

Reviewer #2: Yes

5. Is the manuscript presented in an intelligible fashion and written in standard English?

Reviewer #1: Yes

Reviewer #2: Yes

Reviewer #1: (No Response)

Reviewer #2: After reviewing the revised manuscript, I believe the authors have adequately addressed my comments and made improvements to the content. I have no further suggestions for revision.

**Do you want your identity to be public for this peer review?** For information about this choice, including consent withdrawal, please see our Privacy Policy

Reviewer #1: No

Reviewer #2: No

---

## [Author Response · Author response to Decision Letter 2]

27 Jun 2025

The Reviewers have meticulously examined our manuscript and presented us with a very constructive set of comments and requests, which we believe have improved the strength of our manuscript and the presentation of our work significantly. In the following pages, we address each and every comment. Each section discusses one comment, and refers to the particular point of our revised manuscript that has been amended to accommodate this comment.

We have endeavoured to take all the remarks and requests of the Reviewers strongly into account, and have addressed all comments analytically and very carefully, no matter how minor or major they were. We hope our revision is now up to the standards of PLOS ONE and if requested, we would be glad to address any further issues in subsequent revisions.

---

## [Decision Letter · Decision Letter 2]

21 Nov 2025

Economical Management of Virtual Power Plant Source-network-load-storage in the Context of Electricity-Carbon Market

PONE-D-25-05045R2

Dear Dr. Zhang,

We’re pleased to inform you that your manuscript has been judged scientifically suitable for publication and will be formally accepted for publication once it meets all outstanding technical requirements.

Kind regards,

Baogui Xin, Ph.D.

Academic Editor

PLOS ONE

Additional Editor Comments (optional):

Reviewers' comments:

Reviewer's Responses to Questions

**Comments to the Author**

Reviewer #2: All comments have been addressed

Reviewer #3: (No Response)

Reviewer #4: All comments have been addressed

2. Is the manuscript technically sound, and do the data support the conclusions?

Reviewer #2: Yes

Reviewer #3: Yes

Reviewer #4: Yes

3. Has the statistical analysis been performed appropriately and rigorously?

Reviewer #2: Yes

Reviewer #3: Yes

Reviewer #4: Yes

4. Have the authors made all data underlying the findings in their manuscript fully available?

Reviewer #2: Yes

Reviewer #3: Yes

Reviewer #4: Yes

5. Is the manuscript presented in an intelligible fashion and written in standard English?

Reviewer #2: Yes

Reviewer #3: Yes

Reviewer #4: Yes

Reviewer #2: After a thorough review of the revised version, the authors have effectively addressed the previously raised issues and made improvements in the logical structure and language expression. I find that the paper has met the publication standards and therefore recommend accepting the paper.

Reviewer #3: The paper addresses the challenges of managing Virtual Power Plants (VPPs) that integrate distributed energy resources (wind, solar, storage, EVs, and flexible loads) under the dual-carbon policy. To improve both economic and operational efficiency, the authors propose a cloud–edge–end multi-timescale management framework that coordinates the source–network–load–storage system. The paper is well-written, methodologically sound, and makes a meaningful contribution to VPP management research.

Reviewer #4: This paper proposes a multi-time-scale economic dispatch framework for Virtual Power Plant (VPP) source-network-load-storage in the electricity-carbon market context, utilizing a cloud-edge-end collaborative architecture and a two-layer optimization mechanism. The model is well-structured and supported by sufficient simulation validation, demonstrating theoretical innovation and practical value. The authors have responded thoroughly to the reviewers' comments, with significant improvements in language quality and logical structure. No academic ethics or publication compliance issues were identified. Recommended for acceptance.

**Do you want your identity to be public for this peer review?** For information about this choice, including consent withdrawal, please see our Privacy Policy

Reviewer #2: No

Reviewer #3: No

Reviewer #4: No

---

## [Editor Report · Acceptance letter]

PONE-D-25-05045R2

PLOS ONE

Dear Dr. Zhang,

I'm pleased to inform you that your manuscript has been deemed suitable for publication in PLOS ONE. Congratulations! Your manuscript is now being handed over to our production team.

Kind regards,

on behalf of

Professor Baogui Xin

Academic Editor

PLOS ONE